# Double Repositioning: Veterinary Antiparasitic to Human Anticancer

**DOI:** 10.3390/ijms23084315

**Published:** 2022-04-13

**Authors:** Tania Sultana, Umair Jan, Jeong Ik Lee

**Affiliations:** 1Regenerative Medicine Laboratory, Center for Stem Cell Research, Department of Biomedical Science and Technology, Institute of Biomedical Science and Technology, Konkuk University, Seoul 05029, Korea; tanishabph@konkuk.ac.kr (T.S.); umairjan47@konkuk.ac.kr (U.J.); 2Department of Veterinary Obstetrics and Theriogenology, College of Veterinary Medicine, Konkuk University, Seoul 05029, Korea

**Keywords:** drug repurposing, antiparasitic, benzimidazole carbamates, halogenated salicylanilides, cancer therapy

## Abstract

Drug repositioning, the approach of discovering different uses for existing drugs, has gained enormous popularity in recent years in the anticancer drug discovery field due to the increasing demand for anticancer drugs. Additionally, the repurposing of veterinary antiparasitic drugs for the treatment of cancer is gaining traction, as supported by existing literature. A prominent example is the proposal to implement the use of veterinary antiparasitics such as benzimidazole carbamates and halogenated salicylanilides as novel anticancer drugs. These agents have revealed pronounced anti-tumor activities and gained special attention for “double repositioning”, as they are repurposed for different species and diseases simultaneously, acting via different mechanisms depending on their target. As anticancer agents, these compounds employ several mechanisms, including the inhibition of oncogenic signal transduction pathways of mitochondrial respiration and the inhibition of cellular stress responses. In this review, we summarize and provide valuable information about the experimental, preclinical, and clinical trials of veterinary antiparasitic drugs available for the treatment of various cancers in humans. This review suggests the possibility of new treatment options that could improve the quality of life and outcomes for cancer patients in comparison to the currently used treatments.

## 1. Introduction

Cancer is an extensive disease and the most common cause of morbidity and mortality worldwide. It is characterized by a deregulation of the cell cycle, which primarily results in a progressive loss of control of cellular growth and differentiation [1]. Although there are numerous ongoing studies on anticancer therapy, with many lead candidates at various phases of preclinical or clinical research, only 5% of potential anticancer therapies entering phase I clinical trials have been approved and have entered the market [2]. The standard cancer treatments include surgery, immunotherapy, radiation, and chemotherapy. Currently, chemotherapy is one of the most efficient and potent strategies used to treat malignant tumors. However, the development of multidrug resistance to chemotherapeutics has become a huge impediment to successful cancer treatment. Clearly, new therapeutic alternatives are required to improve cancer diagnosis and treatment. Prior to being marketed as a new drug, the lead compounds face many hurdles during preclinical and clinical studies to ensure their quality, safety, dosage, and efficacy. Clinical trials are costly and time-consuming, requiring ten to fifteen years of dedicated research. The entire development process of getting a single candidate compound onto the market is hindered by the exorbitant costs (approximately $1–$2.5 billion) associated with the necessary trials required for U.S. Food and Drug Administration (FDA) approval [3].

Drug repurposing has gained recognition in the last decade, enabling existing pharmaceutical products to be reconsidered for alternative applications. It has reduced the risk of a drug failing to reach the market, owing to the low burden of adverse effects, the attenuation of the economic load, and the expedition of the approval process [4]. It can also offer an improved risk versus reward trade-off as it shortens the timeline of the drug development process and is also economically feasible when compared to other drug development strategies [5]. Additionally, the preclinical results obtained from the use of repurposed drugs may expedite the process of the preclinical to clinical translation of cancer treatment [6].

Examples of drugs recognized as high-potential agents within the Repurposing Drugs in Oncology project include cimetidine, clarithromycin, diclofenac, mebendazole (MZ) and nitroglycerine, among others [6]. A considerable number of drugs investigated for repurposing in oncology are antiparasitic, and have already been in clinical use against different parasitic worms for several decades [7]. Antiparasitic drugs are a group of candidates that act against the parasitic worms colonizing the intestine. These drugs were originally developed to treat veterinary parasites, and thereafter gained ground for clinical applications in human patients. Potential pharmacological candidates include benzimidazole (BZ) carbamates and halogenated salicylanilides (HS) [8,9], which have been widely used as veterinary antiparasitic drugs and have subsequently been repurposed for human cancer treatment. This laid the ground for the “double repositioning” phenomenon, which can be defined as the repositioning of an existing drug to treat different diseases and species at the same time. The anticancer activities of antihelminthics were reported for MZ, pyrvinium pamoate, and niclosamide (Nic) by Mukhopadhyay et al. in 2002 [10], Esumi et al. in 2004 [11], and Wang et al. in 2009 [12], respectively. Since then, various studies have reported on the effects of these compounds against tumor cells in vitro and in vivo, and in phase 1 clinical trials [6,13]. This review summarizes the current evidence and information on the anticancer activity of BZ and HS group drugs in cell lines, animal tumor models, and clinical trials, which could help improve the quality of life of cancer patients.

## 2. Process of Article Selection

We utilized the search engines PubMed and Google Scholar to gather a list of publications and manuscripts investigating the anticancer activity of BZ carbamate and HS in cell lines, animal tumor models, and clinical trials. For a report to be included in this survey, it must have contained BZ carbamate and HS in either the heading or the abstract. The keywords “BZ carbamate” and “HS”, combined with “albendazole”, “mebendazole”, “fenbendazole”, “flubendazole”, “ricobendazole”, “niclosamide”’ “closantel”, “rafoxanide”, or “anticancer activity” were used to generate the list. Any review article was excluded from our survey. Highly relevant articles were initially determined by the heading and abstract, followed by further examination to confirm whether research conducted with the BZ carbamate and HS groups of drugs was on cell lines or on human or animal subjects. We acknowledge that this search technique was not encyclopedic, as there are many journal’s articles that are not included in PubMed or Google Scholar. We evaluated the chosen studies by assessing different characteristics such as the type of species, the cell source, the cell lines, the cancer type, and the target pathway.

## 3. BZ Carbamates

BZ antiparasitics are a group of heterocyclic aromatic organic compounds that are extensively used in both human and veterinary medicines to inhibit internal parasites. Some important BZ drugs include MZ, albendazole (ABZ), fenbendazole (FZ), flubendazole (FLU), triclabendazole, parbendazole, oxibendazole, and ricobendazole. In the last few years, some of these have been successfully investigated for various types of cancers worldwide.

### 3.1. Mechanism of Action of BZ Carbamates

The molecular mode of action of BZ carbamates involves inhibiting the polymerization of tubulin and facilitating the disruption of microtubules in parasite cells (Figure 1) [14]. An in vitro study using the extracts of helminthic and mammalian tubulin has implicated tubulin as the leading molecular target of BZ carbamates [15]. Tubulin is pivotal to cell motility, proliferation, and division; the intercellular transport of organelles; the maintenance of cell shape; and the secretion process of cells in all living organisms [16]. By blocking microtubule elongation in worms, BZ carbamates perturb glucose uptake in cells. Eventually, the glycogen reserves are exhausted, and their energy management mechanisms are depleted, culminating in the death of the parasites [17].

### 3.2. Anticancer Activity of BZ Carbamates

BZ carbamates are cancer cell-selective, causing minimal cytotoxicity in normal cells but increased cytotoxicity in different tumor cells. Several studies have reported that BZ carbamates inhibit the polymerization of mammalian tubulin in vitro. Whether the same effect would be observed in human cells, and if so, whether such targeted efforts could be effective against tumors, are some questions raised by these reports. Lacey et al. first addressed the activity of BZ carbamates against mouse leukemia cells L1210 in 1985 [18]. A more thorough inquiry into the antitumor effects of BZ carbamates was carried out; the most promising outcomes of this inquiry are summarized in Table 1. The general pharmacokinetic properties of BZ carbamates are as follows: slow absorption; wide distribution throughout the body; extensive hepatic metabolism; and excretion via urine and feces (Figure 2a). Their common side effects are fever, nausea, vomiting, abdominal discomfort, and hepatotoxicity. The low intestinal absorption rate of BZ carbamates may make it difficult for them to reach concentrations in the systemic circulation effective in treating cancers in humans. Increased bioavailability is necessary to enhance their antitumor effect, making them safe and well tolerable in human and veterinary use.

#### 3.2.1. ABZ

ABZ is an FDA-approved drug for the treatment of numerous types of parasitic worm infections, such as cystic hydatid disease of the liver, lung, and peritoneum caused by the larval form of dog tapeworm. It has also been the concern of recent investigations as a significant anticancer agent, due to its limited toxicity to normal cells, but high-level toxicity to both tumor and parasitic cells. In 1989, ABZ cytotoxicity was revealed in the hepatocellular carcinoma (HCC) cell line Hep2G [19]. Additionally, ABZ was found to inhibit the proliferation of human, rat, and mouse HCC cell lines. These in vitro observations were then recapitulated in vivo by the same authors using a xenograft model against the development of subcutaneous human tumor SKHEP-1 cells in nude mice, hinting at the promising antitumor potential of ABZ in humans [20]. ABZ cytotoxicity was examined in different intestinal cancer cell lines (SW480, SW620, Caco2, NCM460, and HCT-8) with different origins and growth characteristics, representing various stages of malignancy [21]. Pougholami et al. provided evidence for its efficacy in the treatment of peritoneal carcinomatosis arising from colorectal cell lines. ABZ and its principal metabolite ABZ sulfoxide were found to markedly inhibit the development and proliferation of the human colorectal cell line HT-29 [22]. The mechanism behind the examined effects of the employed compounds was the induction of cell cycle arrest at the G2/M phase and the activation of caspase-3-mediated apoptosis. Significant cytotoxicity was observed in the paclitaxel (PTX)-resistant leukemia cell-line CEM/dEpoB300 treated with ABZ. ABZ successfully stimulated massive depolymerization of the microtubular network and activation of apoptosis, mediated by a profound downregulation of BCL-2 and MCL-1 proteins [23]. It also facilitated the depolymerization of microtubules in ovarian cancer cells 1A9PTX22, and was found to be more efficacious than PTX in inhibiting their proliferation [24]. ABZ’s low solubility prevents it from being absorbed in large enough quantities in organs to be toxic to humans [101].

#### 3.2.2. FZ

The antitumor activity of FZ was also investigated on different cell lines. Researchers discovered that FZ manifests moderate microtubule depolymerizing activity in human cancer cells, as made evident in in vitro and in vivo experiments. FZ displays its antitumor effect through the disruption of microtubules, the activation of p53, and the modulation of genes associated with multiple cellular pathways. FZ treatment also culminated in reduced glucose uptake in cancer cells by causing the downregulation of GLUT transporters and pivotal glycolytic enzymes [25,26,27,28]. Qiwen et al. reported that FZ treatment was toxic to mammary EMT6 cells in vitro. The toxicity level was augmented with incubation time and under conditions of severe hypoxia. FZ also increased the plastic effects of radiation [26]. It demonstrated a moderate affinity for mammalian tubulin and exhibited cytotoxicity to human cancer cells lines (H460 and A549) at micromolar concentrations. Additionally, it caused the mitochondrial translocation of p53, and efficiently inhibited glucose uptake and the expression of GLUT transporters, as well as hexokinase, a key glycolytic enzyme upregulated in most cancer cells. It also blocked the growth of human xenografts in a *nu/nu* mice model when administered orally. Remarkably, FZ showed no obvious toxicity towards primary epithelial cells from rat lung tissue [27]. However, Ping et al. showed that FZ alone could not affect the growth of the P493-6 human lymphoma cell lines in SCID mice. However, FZ in combination with vitamin supplementation significantly prevented tumor growth, indicating its applicability in antitumor studies [28]. Additionally, the effect of a therapeutic diet containing 150 ppm FZ, injected intradermally into BALB/c mice, on the growth of EMT6 mouse mammary tumors was examined by Duan et al. They concluded that FZ did not alter the tumor growth, invasion, or metastasis, and care should be taken while administering food containing FZ to mice colonies for cancer research [29].

#### 3.2.3. FLU

The anticancer activity of FLU was first investigated in leukemia and myeloma cells originating from both established stabilized cell lines and patient samples. Combinations of FLU with vinblastine or vincristine delayed tumor growth in the xenograft model more than either of the drugs alone. The authors declared FLU as a novel microtubule inhibitor displaying preclinical activity in leukemia and myeloma [30]. In other investigations, FLU successfully inhibited the proliferation of several breast cancer stem-like cell lines (MDA-MB-231, BT-549, SK-BR-3, and MCF-7) in a dose and time-dependent manner, and delayed tumor growth in xenograft models via intraperitoneal injection [31,32,33]. Additionally, FLU showed an anticancer effect against colorectal cancer (CRC) and appeared partly safe to normal cells [34].

#### 3.2.4. MZ

MZ was also found to be effective against many cancer cell lines. MZ significantly inhibited the proliferation of adrenocortical carcinoma cells due to the induction of apoptosis [35]. Additionally, it suppressed the invasion and migration of cancer cells in vitro and metastases formation in experimental animals. The treatment of lung cancer cell lines with MZ caused mitotic arrest by inhibiting the polarization of tubulin followed by apoptosis. The oral administration of MZ in mice stimulated a robust antitumor effect in a subcutaneous model and reduced lung colonies in experimentally induced lung metastasis, without showing any toxicity in PTX-treated mice [10,36]. It is reported that 80% of colon cancer cell lines were sensitive to MBZ in the NCI 60 panel. MBZ caused complete remission of the metastases in the lungs and lymph nodes, with no adverse effect to patients [37,38]. Furthermore, MZ showed anticancer activity against gastric cancer (GC) [39], brain tumor [40], glioblastoma [41], medulloblastoma [42,43], prostate cancer [44], bile duct cancer [45], pancreatic cancer [46], head and neck cancer [47], breast cancer [48], and melanoma [49,50,51]. MZ has been tested preclinically as a replacement for vincristine, a drug that has dose-limiting toxicity in the treatment of brain tumors [102].

### 3.3. Anticancer Activity of BZ Carbamates in Clinical Models

A pilot study of the effect of ABZ in seven patients with advanced hepatocellular carcinoma and CRC with hepatic metastases refractory to other forms of therapy was performed for 28 days. Patients received ABZ orally (10 mg/kg/day) in two divided doses. The levels of tumor markers, carcinoembryonic antigen, and a-fetoprotein were measured routinely. The parameters of hematological and biochemical indices were also obtained to monitor bone marrow, kidney, and liver toxicities. The results of this research further confirm the tolerance for ABZ in patients, with the only side effect of concern being acute neutropenia in three of the patients. Moreover, ABZ significantly reduced two tumor markers in two patients, while in three other patients the markers were stabilized, demonstrating that ABZ possesses antitumor effects in humans [103].

In the subsequent dose-determining phase I trial of oral ABZ in patients with unmanageable solid tumors, 36 patients received an initial 400 mg dose with dose acceleration up to 1200 mg twice a day in three-week cycles. Sequential blood samples were collected for up to 96 h, and on day eight of cycles one and four. The authors concluded that ABZ was well tolerated for the schedule tested in this trial, and that the recommended dose for further studies was 1200 mg twice daily for 14 days in a 21-day cycle [104].

A small-cell lung cancer patient tried FZ with vitamin E supplements, cannabidiol oil, and bioavailable curcumin while going through a clinical trial. A positron emission tomography scan after three months found no cancer cells in the patient’s body [105]. This success story, in addition to 40 other known FZ success stories, was shared through a Social Network Service [106] and a blog [107]. It is also worth mentioning that the research institute Oklahoma Medical Research Foundation, collaborating with Stanford and Emory University, has agreed to assist all cancer patients using FZ with a clinical review of the FZ protocol.

Currently, several clinical trials of MZ on different types of cancers are being conducted. A 48-year-old man with adrenocortical carcinoma ceased taking any chemotherapeutic drug, and was prescribed MZ as a single agent at a dosage of 100 mg twice daily. The cancer metastasis in the patient initially regressed, and subsequently remained stable, showing no clinically significant adverse effects [108]. Another 74-year-old man with metastatic colon cancer was prescribed MZ at a dosage of 100 mg twice daily after other anticancer agents, such as capecitabine, oxaliplatin, and bevacizumab, resulted in intolerable oxaliplatin-induced neuropathy. The patient experienced no adverse effects from the treatment, with nearly complete remission of the metastases in the lungs and lymph nodes and good partial remission in the liver [109]. To investigate the safety and efficacy of individualized dosed MBZ in cancer, 11 patients with advanced gastrointestinal cancer were treated with individualized dose-adjusted MBZ up to 4 g/day to target a serum concentration of 300 ng/mL. Five patients reached the target serum MBZ concentration with no significant adverse effects. Individualized dose-adjusted MBZ was found to be safe and well tolerated in cancer patients with advanced cancer [110]. A phase I open-label research trial on newly diagnosed high-grade glioma patients receiving MZ in combination with temozolomide showed no severe side effects. Patients were administered 500 mg MZ three times a day on a 28-day cycle. The primary goal was to determine the highest tolerated dose of MZ with temozolomide and investigate whether this combination could slow the growth of brain tumors (NCT01729260). The second clinical trial in phases I and II was a pilot study of MZ in combination with carboplatin, temozolomide, and vincristine (NCT01837862). MZ was administered to patients for 70 weeks at a dosage of 100 mg twice a day. The primary aim of phase I was to verify the tolerance of MZ in combination with three current chemotherapeutic agents. Phase II monitored the duration of progression-free status in patients and their overall survival [111]. Furthermore, to evaluate the safety and efficacy of MZ, a phase II clinical trial was carried out in patients with gastrointestinal cancer or cancers of unknown origins (NCT03628079). All patients received continuous treatment with MZ for 16 weeks, and individual doses were determined based on the serum concentration of MZ. The study was terminated in January 2020, with no results reported yet. Additionally, MZ, in combination with metformin, doxycycline, and atorvastatin, has been used on cancer to check the safety, tolerability, and efficacy of combination treatments (NCT02201381). In addition, a Phase 3 adjuvant treatment of MZ used on colon cancer has also been carried out (NCT03925662).

At present, no clinical study on FLU and FZ in human malignancies has been conducted. A more thorough report on clinical trials documenting the antitumor effects of antiparasitic drugs is summarized in Table 2.

## 4. HS

Salicylanilides are a very large group of compounds that show efficient activity against certain types of parasites. Their basic chemical structure consists of a salicylic acid ring and an anilide ring. Examples of HS drugs with potent antihelminthic activity are Nic, rafoxanide (RFX), and closantel.

### 4.1. Mechanism of Action of HS

The primary mechanism of action of HS was investigated in vitro using houseflies and rat liver mitochondria. The authors found an association with the uncoupling of oxidative phosphorylation that halts the production of ATP. This seems to happen through the suppression of the activity of two enzymes, succinate dehydrogenase and fumarate reductase, and thus impairs the motility of parasites and eventually causes death [112]. Several researchers have subsequently confirmed the proposed mechanism in vivo [52,113].

### 4.2. Anticancer Activity of HS

Several HS group drugs have been investigated for their effect on cancer in experimental and preclinical models. The pharmacokinetic properties and common side effects of HS drugs are shown in Figure 2b.

#### 4.2.1. Closantel

Closantel is another halogenated salicylamide agent with potent veterinary antiparasitic activity against internal and external parasites, such as roundworm and liver flukes. It has been reported to inhibit BRAF V600E [53]. Zhu et al. evaluated the antiangiogenic activity of closantel in zebrafish with an IC_50_ of 1.69 μM on the intersegmental vessels and 1.45 μM on the subintestinal vessels [54]. Closantel also remarkably prohibited cancer growth in zebrafish xenotransplanted with human lymphoma, cervical cancer, liver cancer cells, and pancreatic cancer in a dose-dependent manner. They reported that lower concentrations of closantel did not induce any remarkable side effect [54]. However, additional investigations are necessary to confirm the anticancer activity of closantel in humans in order to provide new insights into its possible clinical application.

#### 4.2.2. Nic

Nic is a widely used FDA-approved drug for the treatment of tapeworm infections. Several studies have established the anticancer activity of Nic (5-chloro-N-(2-chloro-4-nitrophenyl)-2-hydrobenzamide) in both in vitro and in vivo models. It has been reported that Nic inhibits the activation and transcriptional function of STAT3 and Wnt/β-catenin signaling by inducing LRP6 degradation effectively, and as a result, it induces the cell growth suppression, cell apoptosis, and cell cycle arrest of cancer cells [55,56]. Moreover, Nic showed no significant toxicity on non-cancerous cells in vitro and revealed no side effects in Nic-treated mice. The inhibitory effects of Nic on cancer stem cells (CSC) anticipate its application for cancer therapy. Nic suppressed the growth of CSC-like cells and mechanistically, with or without cisplatin, significantly inhibited epithelial–mesenchymal transition and tumor growth in breast cancer, with no apparent symptoms of the cytotoxic effects of Nic in the treated mice [57,58,59,60,61]. It has been reported that Nic in combination with TRA-8 suppresses the growth of tumor xenografts in basal-like breast cancers by reducing Wnt/β-catenin activity [62].

Nic has been shown to disrupt the interaction of p65 with FOXM1/β-catenin and inhibit the NF-κB pathway, thus eradicating the leukemia stem cells in chronic myelogenous leukemia [63] and acute myelogenous leukemia [64], respectively. Additionally, Nic inhibited multiple signal transduction pathways, such as the Wnt/β-catenin pathway, and also induced apoptosis in colon cancer, alone or in combination with erlotinib, and may work as a potent therapeutic agent for familial adenomatosis polyposis (FAP) patients by disrupting the Axin–GSK3 complex [65,66,67,68,69,70,71]. Mitochondrial uncouplers such as Nic ethanolamine and oxyclozanide have been found to show significant anticancer activities for treating hepatic metastatic tumors in a mouse model, based on their mitochondrial uncoupling activity [72]. Recently, several investigations have found that Nic prevents the growth of ovarian carcinoma cells, ovarian xenografts, and multiple metabolic signaling pathways, such as Wnt/β-catenin, JAK2/ STAT3, and mTOR signal pathways affecting biogenetics, biogenesis, and redox regulation in ovarian cancer tumor-initiating cells [73,74,75,76,77,78]. The cytotoxicity of Nic was investigated against HCC and human lung cancer cells, where it caused remarkable dose-dependent cell apoptosis, suppressed cell viability, and inhibited clone formation [79,80,81,82,83,84,85]. Furthermore, Nic also showed cytotoxicity in human glioblastoma cells by targeting multiple major cell signaling pathways [86,87,88].

Nic has demonstrated a potential to target the androgen receptor-STAT3 signaling axis, overcoming enzalutamide resistance, as well as suppressing cell migration and invasion in advanced prostate cancer [89,90]. Chen et al. found that Nic remarkably inhibited the mammalian target of rapamycin, the mTOR signaling pathway, and mitochondrial respiration in a panel of cervical cancer cell lines, and suggested Nic as a potential therapeutic drug for cervical cancer [91]. Nic targeted PTX-resistant esophageal cancer cells in vitro and in vivo, via the inhibition of Wnt/β-catenin, whilst showing minimal toxicity towards normal cells [92,93]. It has also been reported that Nic can effectively impede tumor growth in a mouse xenograft tumor model of human osteosarcoma cells by targeting multiple signal transduction pathways [94]. In adrenocortical carcinoma (ACC), Nic exhibits potent anti-ACC activity by inhibiting multiple cellular pathways and damaging the cellular metabolism [95]. Additionally, Nic prohibits the expression of C-MYC and E2F1, while inducing the expression of PTEN in renal cell carcinoma (RCC). Nic has been further shown to synergize with the targeted therapy agent Sorafenib in suppressing RCC cell growth and survival [114]. It has also demonstrated potential efficacy by inhibiting the enrichment of cisplatin-induced human oral squamous cell carcinoma and increasing sensitivity to cisplatin in ALDH+ tumorspheres [115].

#### 4.2.3. RFX

RFX, N-[3-chloro-4-(4-chloro-phenoxy) phenyl]-2-hydroxy-3, 5-diiodobenzamide, is another halogenated salicylanilide compound that is used for the treatment of infections caused by *Fasciola hepatica* and some gastrointestinal roundworms [96,116]. Although RFX is approved by the FDA for veterinary use, little information about its therapeutic effects on humans is available. An early study reported the therapeutic use of RFX in a seven-year-old girl with fascioliasis [117]. Mechanistically, RFX was reported to be a significant B-Raf V600E inhibitor that suppressed the activation of the p38 MAPK pathway, inducing cell apoptosis by reducing the mitochondrial membrane potential in multiple myeloma [53,97]. RFX exhibited no toxicity on two human cell lines at concentrations up to 12.5 μM in a study evaluating the effects of RFX on the hypervirulent stationary-phase of *Clostridium difficile* [98], and it significantly lowered hemolytic activity compared to Nic [98].

Shi et al. investigated the inhibition of CDK4/6 activity by RFX, and suggested it as a promising candidate for the treatment of human skin cancer. The study did not observe any noticeable toxicity or change in the body weight of the BALB/C nude mice administered with REF [99]. Furthermore, the systemic administration of RFX to mice induced apoptosis and accelerated the inhibition of colon carcinogenic cells by activating the ERS cascade [100]. More recently, studies have indicated that RFX is a bona fide immunogenic cell death (ICD) inducer in CRC cells. Broadly, it induced all the main damage-associated molecular patterns (ecto-calreticulin exposure and ATP/high mobility group box 1 release) essential for ICD. In vivo, RFX remarkably reduced tumor proliferation in CRC HCT-116 and DLD1 CRC cells compared to sham [118]. RFX’s anticancer effects were also revealed in GC, where it suppressed the proliferation of GC cells in the G0/G1 phase of the cell cycle and stimulated apoptosis and autophagy through the inhibition of the PI3K/Akt pathway both in vitro and in vivo [119].

### 4.3. Anticancer Activity of HS in Clinical Models

Nic underwent clinical trials in patients with resectable colon cancer in 2017, but was terminated because of the low enrolment rate (NCT02687009). Two other clinical studies are currently underway to test the anticancer effects of Nic in patients with FAP (NCT04296851) and progression of metastases of colorectal cancer after therapy (NCT02519582). Although a phase I trial of Nic administered together with enzalutamide in patients with castration-resistant prostate cancer has concluded, anticipating the commencement of a phase 2 trial (NCT02532114), another phase I clinical trial is investigating the potent dose and side effects of Nic in combination with enzalutamide to treat castration-resistant prostate cancer patients (NCT03123978). A phase II clinical trial is also ongoing to evaluate the efficacy of abiraterone acetate, Nic, and prednisone in treating patients with hormone-resistant prostate cancer (NCT02807805).

## 5. Discussion

The repurposing of approved and abandoned drugs has gained enormous attention in oncology drug development, representing a solution to the exorbitant cost of discovering new drugs. Furthermore, approval may be expedited by knowledge of preclinical and clinical data on pharmacokinetics, toxicities, and regimens [120]. There is an interest in the anticancer activities of several antiparasitic agents due to their less severe side effects compared to conventional cancer chemotherapeutics (mainly DNA damaging agents) on the quality of patients’ life. BZ carbamates and HS drugs, when administered alone or in combination, elicit anti-tumor activities demonstrated by various biological actions, such as inducing apoptosis and autophagy; reducing cell viability, migration, and invasion; disrupting tubulin polymerization; inducing differentiation and senescence; reducing angiogenesis; impairing glucose utilization; arresting the cell cycle; and targeting several key oncogenic signal transduction pathways. These agents induce minimal cytotoxicity in normal cells, but comparatively high cytotoxicity in tumor cells, exerting cancer cell-specific selectivity. In addition to their significant effect on cancer cell lines, BZ carbamates and HS drugs have shown antitumor effects in in vivo animal cancer cell models, including reduced tumor growth and vessel formation, reduced metastasis, prolonged overall survival, and progression-free survival.

However, clinical trials remain the major bottleneck for the successful repositioning of veterinary antiparasitic agents in oncology because sufficient and useful data are not always forthcoming. Another impediment to the successful repositioning of veterinary antiparasitic drugs as anticancer agents is related to their physicochemical properties, route of administration, and poor bioavailability. Although the safety of BZ carbamates and HS drugs for veterinary care has been assessed, their use as anticancer agents has not been well established for human application. The potential activities, such as toxicity, side effects, and efficacy, of the repurposed drugs on cancer and non-cancer cell populations following both short and long-term administration warrant in-depth experimental investigations in the future.

Despite limited data, some of these drugs have already been under phase II/III clinical trials for cancer therapy. Some patients with metastatic late-stage cancers responded well to ABZ, MZ, and Nic, showing diminished tumor markers and metastasis and stabilized disease progression. Ongoing clinical trials will provide further clarity on the potential of BZ carbamates and HS drugs for use in primary and metastatic cancer treatments in the near future. There is also evidence that many desperate patients are taking antiparasitic drugs without informing their physicians [107]. When a patient self-prescribes antiparasitic drugs in combination with an established anticancer drug during a clinical trial, the outcome of the clinical trial can be altered; this may result in huge economic losses and the squandering of precious time. In such scenarios, the reasons for the therapeutic effects or the adverse effects of the trial drug in the clinical trial will remain equivocal. The cancer specialist will be confounded as a result of not knowing of the hidden administration of the self-prescribed antiparasitic drug. To prevent this, clinical trials of antiparasitic drugs need to be prioritized. Information about the evidence of antitumor activity and the absence of toxicity in the long-term administration of these drugs can smoothen the path for future clinical trials and determine the best dosage regime.

Therefore, we argue in favor of these compounds, given their well-established pharmacokinetics, excellent toxicity profile, and low cost. We believe a primary interview of cancer patients along with interdisciplinary researchers such as veterinarians, cancer specialists, and pharmacologist, will be worthwhile to facilitate clinical trials of veterinary antiparasitic drugs with potential use for cancer patients.

## Figures and Tables

**Figure 1 ijms-23-04315-f001:**
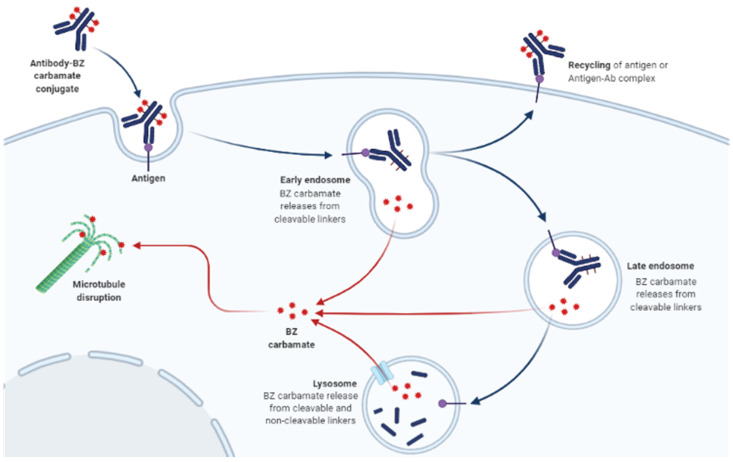
Mechanism of action of benzimidazole (BZ) carbamates targeting tubulin. Tubulin is the leading molecular target of BZ carbamates. They selectively bind to parasitic β-tubulin, promoting their immobilization and death. dapted from “Antibody-Drug Conjugate Drug Release”, by BioRender.com (2022). Retrieved from https://app.biorender.com/biorender-templates, accessed on 10 March 2022.

**Figure 2 ijms-23-04315-f002:**
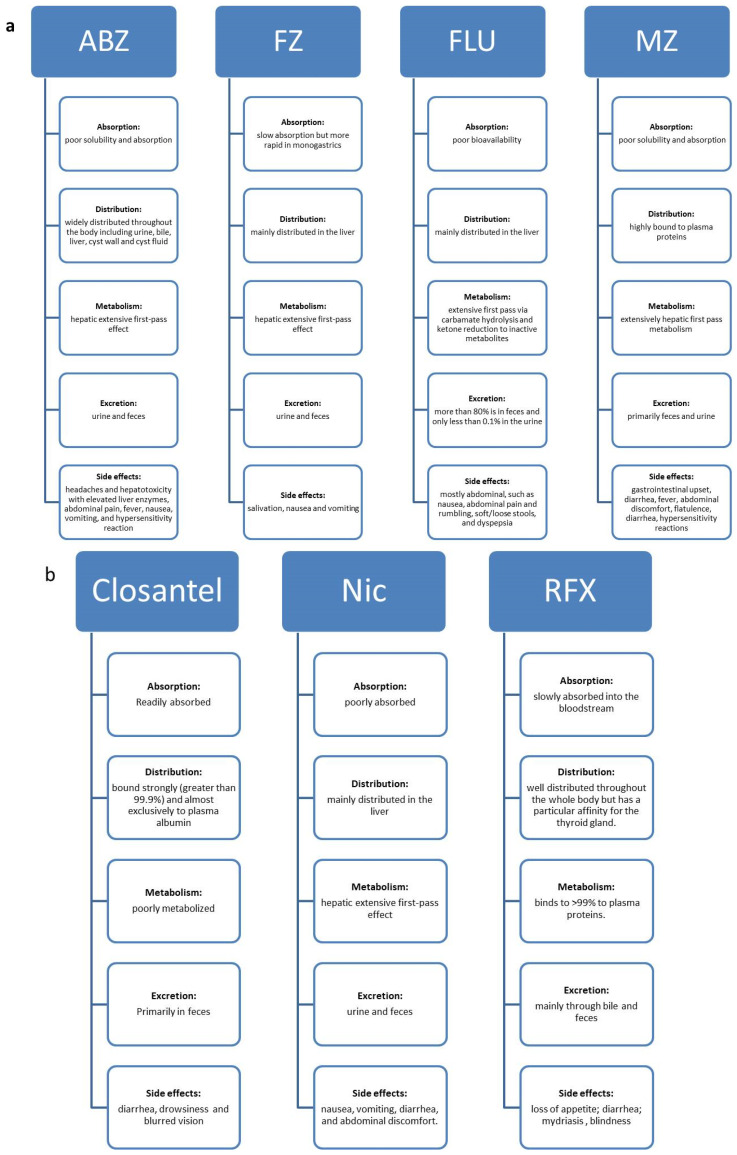
Pharmacokinetic properties and side effects of veterinary antiparasitic drugs. (**a**) The BZ carbamate drugs are poorly absorbed; have a wide distribution in the body; show extensive hepatic metabolism; and are excreted via feces and urine. (**b**) The halogenated salicylanilides (HS) antiparasitic drugs show poor absorption, distribution throughout the body, are poorly metabolized and are excreted in bile, feces, or urine.

**Table 1 ijms-23-04315-t001:** Anticancer activity of BZ carbamates.

Cell Source	Cell Lines	Procedure of Study	Species	Antiparasitics	Cancer Type	Target Pathway	Reference
Human	Hep G2 and Hep3B	in vitro	*Mice*	ABZ	HCC	Cytotoxicity	[19]
Human	Hep G2 and Hep3B, PLC/PRF/5 and SKHEP-1	in vitro	*Mice*	ABZ	HCC	Tubulin disruption	[20]
	SKHEP-1	in vivo					
Rat	HTC, Novikoff	in vitro					
Mice	Hep1-6	in vitro					
Human	SW480, SW620, HCT8 and Caco2	in vitro	*Mice*	ABZ, RBZ, FLU	Intestinal cancer	Tubulin disruption	[21]
Human	HT-29	in vitro	*Mice*	ABZ	CRC	Apoptosis	[22]
Human	CEM/dEpoB300	in vitro	*Mice*	ABZ	Leukemia	Apoptosis	[23]
Human	1A9Pc TX22	in vitro	*Mice*	ABZ	OC	Apoptosis	[24]
Mouse	EMT6	in vitro	*Mice*	FZ	Mammary carcinoma	Cytotoxicity	[25]
		in vivo					
Human	H460 and A549	in vitro	*nu/nu* *mice*	FZ	LC	microtubule disruption, p53 activation and down regulation of pivotal glycolytic enzymes	[26]
		in vivo					
Human	P493-6	in vitro	*SCID mice*	FZ	Lymphoma	Tubulin disruption	[27]
		in vivo					
Mice	EMT6	in vitro	*BALB/c Rw mice*	FZ	Mammary carcinoma	Tubulin disruption	[28]
		in vivo					
Human	OCI-AML-2	in vitro	*SCID mice*	FLU	Leukaemia and Myeloma	Tubulin disruption	[29]
		in vivo					
Human	MDA-MB-231, BT-549, SK-BR-3 and MCF-7	in vitro	*Mice*	FLU	BC	Tubulin disruption	[30]
		in vivo					
Human	TNBC cell lines MDA-MB-231 and MDA-MB-468	in vitro	*Mice*	FLU	BC	Apoptosis	[31]
		in vivo					
Human	BT474, SK-BR-3, MDA-MB-453, JIMT-1	in vitro	*BALB/c mice*	FLU	BC	Tubulin disruption	[32]
		in vivo				Apoptosis	
Human	HCT116, RKO and SW480	in vitro	*BALB/c mice*	FLU	CRC	Apoptosis	[33]
		in vivo					
Human	H295R and SW-13	in vitro	*Mice*	MZ	Adrenocortical carcinoma	Apoptosis	[34]
Human	H460, A549, H1299 and WI-38	in vitro	*Mice*	MZ	LC	Tubulin disruption, Apoptosis	[35]
		in vivo					
Human	HCT 116 and RKO	in silico	*-*	MZ	CC	Tubulin disruption	[36]
Human	DLD-1, HCT-116, HT-29 and SW480	in vitro	*Mice*	MZ	CC	Tubulin disruption	[37]
Human	ACP-02, ACP-03 and AGP-01	in vitro	*Mice*	MZ	GC	Tubulin disruption	[38]
		in vivo					
Mouse	GL261	in vitro	*C57BL6 Mice*	MZ	Brain tumour	Tubulin disruption	[39]
		in vivo				Apoptosis	
Human	GBM U87-MG, D54, H80, H247, H392, H397, H502 and H566	in vitro	*C57BL/6 mice*	MZ	Brain cancer	Apoptosis	[40]
		in vivo					
Mouse	GL261						
Human	D425 MB	in vivo	*p53 mice*	MZ	Medullo-blastoma	Tubulin disruption	[41]
Human	293T and hTERT-RPE1	in vitro	*nu/nu* athymic mice	MZ	Medullo-blastoma	Hedgehog inhibitor	[42]
		in vivo					
Murine	CP2 and SP1	in vitro	*BALB/c mice*	MZ	PC	Tubulin disruption	[43]
		in vivo					
Human	KKU-M213	in vitro	*Nude mice*	MZ	Bile duct Cancer	Apoptosis	[44]
		in vivo					
Human	PANC-1	in vitro	*Mice*	MZ	Pancreatic cancer	-	[45]
Human	CAL27 and HCC15	in vitro	*Nude mice*	MZ	Head and neck cancer	Apoptosis	[46]
		in vivo					
Human	SK-Br-3	in vivo	*Mice*	MZ	BC	Tubulin disruption	[47]
Human	M-14 and SK-Mel-19	in vitro	*Mice*	MZ	Melanoma	Tubulin disruption	[48]
Human	MM622, MM540, D08, MM329, D17, and UACC1097	in vitro	*Mice*	MZ	Melanoma	Tubulin disruption	[49]
		in vivo					
Human	NRAS^Q61K^	in vitro	*Athymic mice*	MZ	Melanoma	Apoptosis	[50]
		in vivo					
		in silico					
Human	GL261	in vitro	*C57BL/6 mice*	MZ	Brain cancer	Tubulin disruption	[51]
		in vivo					
Human	Burkitt’s lymphoma Ramos cells, Hela cells, PANC-1 cells, and HepG2 cells	in vivo	*Zebra-fish*	Closantel	Lymphoma, cervical cancer, PC, and LC	Suppression of antiangiogenesis and Closantel	[52]
Human	Du146	in vitro	*Mice*	Nic	PC	Inhibition of STAT3 Pathway	[53]
Human	HEK293 cells	in vitro	*Mice*	Nic	PC and BC	Inhibition of Wnt/β-catenin Pathway	[54]
Human	MCF7 and MDA-MB-231	in vitro	*NOD/SCID mice*	Nic	BC	Apoptosis and downregulation stem pathways	[55]
		in vivo					
Human	MDA-MB-231	in vitro	*BALB/c nude mice*	Nic and cisplatin	BC	Apoptosis and inhibition of Akt, ERK, and Src pathways	[56]
		in vivo					
Human	MDA-MB-468 and MCF-7	in vitro	*Mice*	Nic	BC	Inhibition of cell motility and STAT3 activity	[57]
Human	TNBC MDA-MB-231, MDA-MB-468 and Hs578T	in vitro	*Athymic nude mice*	Nic	BC	Inhibition of Wnt/β-catenin Pathway	[58]
		in vivo					
Mouse	4T1	in vitro	*BALB/c mice*	Nic	BC	Apoptosis and suppression of cell migration and invasion	[59]
		in vivo					
Human	MDA-MB-231, MDA-MB-468 and MCF-7	in vitro					
Human	2LMP, SUM159, HCC1187, and HCC1143	in vitro	*NOD/* *SCID mice*	Nic	BC	Cytotoxicity	[60]
		in vivo					
Human	K562 and KBM5-T315I cells	in vitro	*NOD mice*	Nic	Chronic myelogenous leukemia	Inhibition of FOXM1/β-catenin Pathway	[61]
		in vivo					
Human	HL-60, U937, OCI-AML3, Molm13, MV4-11, and U266 cells	in vitro	*BALB/c mice*	Nic	Acute myelogenous leukemia	Apoptosis and Inhibition of NF-κB pathway	[62]
		in vivo					
Human	MCF7	in vitro	*Mice*	Nic	Adeno-carcinoma	Inhibition of PI3K-dependent signalling	[63]
	HCC1954				Carcinoma		
	BT-474	in vivo			Ductal Carcinoma		
	MDA-MB-361 and				Adeno-carcinoma		
	SKBR3 cell	in silico			Adeno-carcinoma		
Human	HCT116, SW620, and HT29	in vivo	*Mice*	Nic	CC	Inhibition of STAT3 phosphorylation	[64]
Human	HCT116, SW480, DLD1 and 293 cells	in vitro	*APC-MIN mice*	Nic	CC	Inhibition of Wnt/Snail-mediated EMT	[65]
		in vivo					
Human	HCT116, SW620, LS174T, SW480, and DLD-1	in vitro	*NOD/SCID mice*	Nic	CC	Inhibition of S100A4-induced metastasis formation	[66]
		in vivo					
		in situ					
Human	HT29, HCT116, CaCO2 and MCF-10A	in vitro	*NOD/SCID mice*	Nic	CC	Inhibition of Wnt/β-catenin Pathway	[67]
		in vivo					
Human	HEK293T, U2OS, WIDR, DLD-1, CRC 240, COLO205, CRC57 and HCT116	in vitro	*Mice*	Nic	CC	Induction of autophagy and inhibition of Wnt/β-catenin Pathway	[68]
Human	SW480 and SW620	in vitro	*Mice*	Nic	CC	Reduction of Wnt activity	[69]
Rodent	CC531	in vivo					
Murine	MC38	in vitro	*APC^min/+^ mouse*	Nic-EN and oxyclozanide	CC	Mitochondrial uncoupling	[70]
		in vivo					
Human	HCT116	in vitro					
Rodent	C2C12	in vitro					
		in vivo					
Human	SKOV3 and CP70	in vitro	*SCID mice*	Nic	OC	Induction of metabolic shift to glycolysis	[71]
		in vivo					
Human	OVCAR-3, SKOV-3 and A2780	in vitro	*NOD/* *SCID mice*	Nic	OC	Inhibition of CP70sps and primary OTICs	[72]
		in vivo					
Human	SKOV3.ip1	in vitro	*Mice*	Nic	OC	Inhibition of Wnt/β-catenin Pathway	[73]
		in vivo					
Human	SKOV3 and HO8910	in vitro	*Athymic Nude mice*	Nic	OC	Mitochondrial Respiration and aerobic glycolysis	[74]
		in vivo					
Human	A2780ip2, A2780cp20, and SKOV3Trip2	in vitro	*SCID* *mice*	Nic	OC	Inhibition of Wnt/β-catenin, mTOR and STAT3 pathways	[75]
		in vivo					
Human	Tumorspheres	in vitro	*Mice*	Nic and its analogs in combination with carboplatin	OC	Cytotoxicity	[76]
		in vivo					
Human	HepG2 and QGY7701	in vitro	*Mice*	Nic	HCC	Apoptosis and suppression of ATF3 expression	[77]
Human	NSCLC, NCI-H1299 and HCT116	in vitro	*Mice*	Nic	LC	Apoptosis through ROS-mediated p38 MAPK-c-Jun activation	[78]
Human	SK-Hep-1 and Huh7	in vitro	*Mice*	Nic	HCC	Inhibition of metastasis of HCC, and CD10	[79]
Human	HCC827, H1650, and H1975	in vitro	*Nu/Nu nude mice*	Nic	LC	Inhibition of STAT3 phosphorylation	[80]
		in vivo					
Human	A549/DDP	in vitro	*Mice*	Nic combined with cisplatin (DDP)	Cisplatin-resistant LC	Apoptosis and reduction of c-myc protein	[81]
Human	HepG2, QGY-7703 and SMMC-7721	in vitro	*Mice*	Nic	HCC	Inhibition of cell growth and STAT3 pathway	[82]
Human	Lung adenocarcinoma (549, EKVX, H358, Hop62, H322M, H522, H838, and H23), large cell lung carcinoma (H460, Hop92), NCSLC (H1299, H810) and small cell LC (H82)	in vitro	*Mice*	Nic	LC	Reduction in proliferation and inhibition of S100A4 protein	[83]
Human	U-87 MG	in vitro	*Mice*	Nic	Glioblastoma	Cell toxicity and inhibition of Wnt/β-catenin, PI3K/AKT, MAPK/ERK, and STAT3	[84]
Human	TS15-88, GSC11	in vitro	*Athymic nude mice*	Nic and/or temo-zolomide	Glioblastoma	Inhibition of the expression of epithelial-mesenchymal transition-related markers, Zeb1, N-cadherin, and β-catenin	[85]
		in vivo					
Human	LN229, T98G, U87(MG), U138, and U373(MG)	in vitro	*Rag2^−/−^Il2rg^−/−^ and SCID/* *Beige mice*	Nic	Glioblastoma	Cytotoxicity and diminished the pGBMs’ malignant potential	[86]
		in vivo					
Human	C4-2B, LNCaP and DU145	in vitro	*Mice*	Nic with enzalutamide	Enzalutamide resistance PC	Inhibition of migration, invasion and IL6-Stat3-AR pathway	[87]
Human	LNCaP, VcaP, CWR22Rv1, PC3 and HEK293	in vitro	*SCID mice*	Nic with enzalutamide	Castration-resistant PC	Inhibition of AR variant and enzalutamide-resistant tumor growth	[88]
		in vivo					
Human	CaLo, HeLa, SiHa, CasKi, DoTc2, ViBo and C-33A	in vitro	*SCID mice*	Nic	Cervical cancer	Inhibition of mTOR signaling	[89]
		in vivo					
Human	ESO26, FLO-1, KYAE-1, OE33, SK-GT-4, and OE19	in vitro	*SCID mice*	Nic	Esophageal cancer	Inhibition of Wnt/β-catenin	[90]
		in vivo					
Human	BE3,CE48T/VGH and CE81T/VGH	in vitro	*Mice*	Nic	Esophageal cancer	Inhibition of cell proliferation and STAT3 pathway	[91]
Human	Osteosarcoma cells	in vitro	*Mouse*	Nic	Osteosarcoma	Apoptosis and target multiple signaling pathways	[92]
		in vivo					
Human	NCI-H295R and SW-13	in vitro	*Nu^+^/Nu^+^ mice*	Nic	Adrenocortical Carcinoma	Induction of G_1_ cell-cycle arrest mitochondrial uncoupling	[93]
		in vivo					
Human	A498 and Caki-1	in vitro	*Athymic nude mice*	Nic	Renal cell carcinoma	Inhibition of cell proliferation, migration and cell cycle progression	[94]
		in vivo					
Human	SCC4 and SCC25	in vitro	*Mice*	Nic	Oral cancer	Inhibition of cancer stemness, extracellular matrix remodeling, and metastasis through dysregulation Wnt/β-catenin signaling pathway	[95]
Human	H929, MM1S, U266 and BMSC	in vitro	*BALB/c nude mice*	RFX	Multiple myeloma	Apoptosis and inhibition of DNA synthesis	[96]
		in vivo					
Human	A431 and A375	in vitro	*BALB/c nude mice*	RFX	Skin cancer	Inhibition of CDK4/6	[97]
		in vivo					
Human	HCT-116 and HT-29	in vitro	*Apc*^min^*/+* mice	RFX	CRC	Inhibition of cell proliferation	[98]
		in vivo					
Human	HCT-116 and DLD1 cells	in vitro	*BALB/c nude mice*	RFX	CRC	Induction of ICD of CRC cells	[99]
		in vivo					
Human	SGC-7901 and BGC-823, GES-1	in vitro	*BALB/c nude mice*	RFX	GC	Apoptosis and inhibition of PI3K/Akt/mTOR signaling pathway	[100]
		in vivo					

PubMed, Google Scholar, and CTD databases were used to summarize the data for the antitumor effects of BZ carbamates. ABZ—albendazole; BC—breast cancer; CC—colon cancer; CRC—colorectal cancer; EMT—epithelial–mesenchymal transition; FZ—fenbendazole; GC—gastric cancer; HCC—hepatocellular carcinoma; ICD—immunogenic cell death; LC—lung cancer; MZ—Mebendazole; Nic—Niclosamide; Nic-EN—Niclosamide ethanolamine; OC—ovarian cancer; PC—prostate cancer; RBZ—Ricobendazole; RFX—Rafoxanide.

**Table 2 ijms-23-04315-t002:** Application of veterinary antiparasitic drugs in clinical trials used to treat different types of cancers.

Antiparasitics	Cancer Type	Title	Phase	Purpose	Status/Result	Identifier/Ref
ABZ	HCC or CRC	Pilot Study Of Albendazole In Patients With Advanced Malignancy. Effect On Serum Tumor Markers/High Incidence Of Neutropenia	PS	Evaluation of anticancer activity	Stabilization of the disease, but because of neutropenia, treatment was stopped on day 19	[102]
ABZ	Refractory solid tumors	Phase I Clinical Trial To Determine Maximum Tolerated Dose Of Oral Albendazole In Patients With Advanced Cancer	1	To determine the safety, tolerability, and the maximal tolerated dose. To characterize the pharmacokinetics and preliminary evidence of efficacy.	2400 mg/day from 1200 mg b.d. Decreased plasma VEGF and 16% patients had a tumor marker response with a fall of at least 50%	[103]
MZ	Adreno-cortical carcinoma	Mebendazole Monotherapy and Long-Term Disease Control in Metastatic Adrenocortical Carcinoma	CS	To describe successful long-term tumor control	Well tolerated, and the associated adverse effects of MZ are minor	[107]
MZ	CC	Drug Repositioning From Bench To Bedside: Tumour Remission By The Antihelmintic Drug Mebendazole In Refractory Metastatic Colon Cancer	CS	Repositioning drugs for use in advanced CC	No disease-related symptoms were found	[108]
MZ	Glio- blastoma	Mebendazole In Newly Diagnosed High-Grade Glioma Patients Receiving Temozolomide (Mebendazole)	1	To find the highest dose and the efficiency of MZ to slow the growth of the brain tumor	Active, not recruiting	NCT01729260
MZ	Pediatric Gliomas	A Phase I Study of Mebendazole for the Treatment of Pediatric Gliomas	1	To determine the safety and efficacy of MZ	Recruiting	NCT01837862
MZ	GI Cancer	A Clinical Safety and Efficacy Study of Mebendazole on GI Cancer or Cancer of Unknown Origin. (RepoMeb)	1	To determine the safety and efficacy of MZ (ReposMZ)	Terminated (Lack of effect)	NCT03628079
	Cancer of Unknown Origin		2			
MZ	OC, PC and ovarian epithelial cancer	Study of the Safety, Tolerability and Efficacy of Metabolic Combination Treatments on Cancer (METRICS)	3	To determine the effectiveness of a regimen of selected metabolic treatments for cancer patients and to perform exploratory analysis on the relationship between the degree of response and changes in biochemical markers	Not yet recruiting	NCT02201381
MZ	CC	Mebendazole as Adjuvant Treatment for Colon Cancer	3	MZ as adjuvant treatment for colon cancer	Recruiting	NCT03925662
Nic	CC	A Study of Niclosamide in Patients With Resectable Colon Cancer	1	To determine the maximum tolerated dose (MTD)	Terminated (low accrual)	NCT02687009
Nic	CRC	Drug Trial to Investigate the Safety and Efficacy of Niclosamide Tablets in Patients With Metastases of a Colorectal Cancer Progressing After Therapy (Nikolo)	2	To evaluate the safety and efficacy of oral appliqued Nic	Unknown	NCT02519582
Nic	PC	Niclosamide and Enzalutamide in Treating Patients With Castration-Resistant, Metastatic PC	1	To determine the side effects and best dose of Nic	Completed (No result posted)	NCT02532114
Nic	Metastatic PC	Enzalutamide and Niclosamide in Treating Patients With Recurrent or Metastatic Castration-Resistant PC	1	To determine the best dose and side effects of Nic when given together with enzalutamide	Recruiting	NCT03123978
	Recurrent PC					
	Stage IV PC					
Nic	Metastatic PC	Abiraterone Acetate, Niclosamide, and Prednisone in Treating Patients With Hormone-Resistant PC	2	To determine the side effects and how well abiraterone acetate, Nic, and prednisone work in treating patients with hormone-resistant PC	Recruiting	NCT02807805
	Recurrent PC					
	Stage IV PC					

NCBI database was used to inquire about the clinical trials on antitumor effects of antiparasitic drugs. ABZ—albendazole; MZ—mebendazole, Nic—niclosamide, CS—clinical study; CC—colon cancer; CRC—colorectal cancer; HCC—hepatocellular carcinoma; PS—pilot Study.

## Data Availability

Not applicable.

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
