# Peer review of "Double Repositioning: Veterinary Antiparasitic to Human Anticancer"

_ijms, 2022, doi:10.3390/ijms23084315_

Round 1
Reviewer 1 Report
The manuscript is based on research in the scientific literature and presents information on experimental, preclinical and clinical studies of veterinary antiparasitic drugs for the treatment of various types of human cancer. This review will help improve the quality of life in cancer patients, suggesting the possibility of new treatment options compared to the conventional one. Potential pharmacological candidates for research are benzimidazole carbamates (BZ) and halogenated salicylanilide (HS) drugs, which were initially widely used as veterinary antiparasitic drugs and subsequently reused for human cancer. These studies initiated the phenomenon of "double repositioning", which can be defined as the repositioning of an existing drug for the treatment of different diseases and species at the same time. In general, the quality of the article is good and, overall, the manuscript is interesting to readers. English language and style are good, but there are some minor spelling mistakes. In conclusion, I consider the article could be a useful contribution to the journal. I recommend the manuscript for being published.
Author Response
Dear Reviewer,
Thank you for finding merit in our manuscript for cancer treatment. We appreciate your time and critical evaluation. As per your suggestion, the manuscript has undergone professional English editing. We believe that you will find the revised manuscript suitable for publication.
Regards
Authors Team

Reviewer 2 Report
The review manuscript by Sultana et al. presents preclinical and clinical updates on veterinary antiparasitic as promising non-oncology drugs that could be repurposed for the treatment of several malignancies. The topic is hot and interesting, but there are several concerns that need to be addressed, or there are missing contents that require a discussion.
The review lacks additional and relevant references on the antitumor potential of benzimidazole anthelmintics and both English language and style require extensive editing, due to the presence of many inconsistencies and typos.
Throughout the text, when referring to tumor cell lines, there is often no indication of what type of tumor it is. Moreover, too many case reports on single patients are mentioned (generally, more robust and consistent data need to be enclosed in a review article).
The review lacks at least one Figure supported by mechanistic data reported in the main text, and also Tables 1 and 2 need to be improved in contents, editing, language and headings (for example, in Table 1 it is unclear whether studies are conducted in vitro or in vivo, since cell lines and “species” are reported in the same row; moreover, in “Cancer type” column, “cancer” does not make sense, and so on; Table 2 needs to be reorganized, avoiding studies completed without results or recruiting).
Overall, I think that the manuscript needs to be substantially improved and reconsidered after major revision.
Author Response
Dear Reviewer,
Thank you for finding our manuscript interesting and beneficial for cancer treatment. We have revised the manuscript according to your suggestions as much as possible.
The review manuscript by Sultana et al. presents preclinical and clinical updates on veterinary antiparasitic as promising non-oncology drugs that could be repurposed for the treatment of several malignancies. The topic is hot and interesting, but there are several concerns that need to be addressed, or there are missing contents that require a discussion.
The review lacks additional and relevant references on the antitumor potential of benzimidazole anthelmintics and both English language and style require extensive editing, due to the presence of many inconsistencies and typos.
- Thank you for your comment. We have listed more than 80 references indicating the antitumor effect of benzimidazole carbamate conducted in vivo and in vitro using the search engine PubMed and Google Scholar. We have also included a table listing the anticancer therapies of different benzimidazole carbamates at various phases of preclinical or clinical research. However, we give credence that this search technique was not encyclopedic, as there are many journals articles that are not included in PubMed or Google Scholar. We evaluated the chosen studies by assessing different characteristics such as the type of species, cell source, cell lines, cancer type, and target pathway.
- Thank you for your comment. As per your suggestion, our manuscript has undergone professional English editing. We have submitted the certificate from editing system as well during this submission. This time, we believe that you will find the revised manuscript suitable for publication.
Throughout the text, when referring to tumor cell lines, there is often no indication of what type of tumor it is. Moreover, too many case reports on single patients are mentioned (generally, more robust and consistent data need to be enclosed in a review article).
- Thank you for your comment. We have included the name of the cell lines and the type of tumors in the manuscript wherever applicable. The changes and revisions are marked in yellow, in the manuscript.
- Thank you for your comment. We have only mentioned two case studies in our manuscript, which we believe are not ‘too’ many. However, as per your suggestion, we have added new case study with large number patients highlighted in yellow.
The review lacks at least one Figure supported by mechanistic data reported in the main text, and also Tables 1 and 2 need to be improved in contents, editing, language and headings (for example, in Table 1 it is unclear whether studies are conducted in vitro or in vivo, since cell lines and “species” are reported in the same row; moreover, in “Cancer type” column, “cancer” does not make sense, and so on; Table 2 needs to be reorganized, avoiding studies completed without results or recruiting).
- Thank you for bringing this to our attention. We have added two figures indicating the mechanism of action of benzimidazole carbamate and the pharmacokinetic properties of veterinary antiparasitic drugs.
- Thank you for your comment. We have made appropriate corrections in Tables 1 and 2 with the help of a professional English Editing Service. We have also added an additional column in Table 1 specifying the procedure of the experiment (in vivo/in vitro) and also specified the type of cancer instead of writing only ‘Cancer.’
- Thank you for your comment. We have reorganized Table 3 by including additional details. However, we believe that the clinical trials, which are still recruiting, are worth mentioning as they show that antiparasitic drugs have a high likelihood of being used clinically in the future.
Overall, I think that the manuscript needs to be substantially improved and reconsidered after major revision.
We thank you again for taking the time and effort to critically evaluate our manuscript. We hope you find our revised manuscript suitable for publication.
Regards
Authors Team

Reviewer 3 Report
This reviewer understands that the title does not represent the type of review performed. It must be changed.
There is a lack of image data or graphics that relate to the pharmacological profiles of drugs.
Points 4.1, 4.2 and 4.3 need to be developed.
There is not enough complete information about drugs that are exclusively veterinary antiparasitic and start to be applied to human cells.
Data on the results in normal human cells (toxicity assessment) of these drugs (veterinary antiparasitic) are lacking.
The manuscript must be reformulated for later resubmission.
Author Response
Dear Reviewer,
Thank you for finding our manuscript of interest for cancer treatment. We have made suitable modifications in the manuscript according to your suggestions to the best of our abilities.
This reviewer understands that the title does not represent the type of review performed. It must be changed.
- Thank you for your comment. We have changed the title to “Potentiality of Veterinary Antiparasitics as Human Anticancer Agents.” However, we welcome your suggestions as well.
There is a lack of image data or graphics that relate to the pharmacological profiles of drugs.
- Thank you for bringing this to our attention. We have added two figures (highlighted in cyan) indicating the mechanism of action of benzimidazole carbamate and the pharmacokinetic properties of veterinary antiparasitic drugs.
Points 4.1, 4.2 and 4.3 need to be developed.
- Thank you for your comment. We have improved the grammar and English of the paper and have rechecked sections 4.1, 4.2, and 4.3 and corrected any minor errors noted and updated the contents, as needed. The changes are highlighted in cyan and green.
There is not enough complete information about drugs that are exclusively veterinary antiparasitic and start to be applied to human cells.
- Thank you for your comment. We listed the references mentioning selected antiparasitics and their antitumor effect along with cancer drugs exhibiting similar anticancer properties in humans.
Data on the results in normal human cells (toxicity assessment) of these drugs (veterinary antiparasitic) are lacking.
- Thank for your comment. We have added the description of toxicity of selected drugs in the manuscript (highlighted in gray). In addition, we have added the side effects of antiparasitic drugs in Figure 2.
The manuscript must be reformulated for later resubmission.
We thank you again for taking the time and effort to critically evaluate our manuscript. We have changed the manuscript according to your suggestions. We hope you find our revised manuscript suitable for publication.
Regards
Authors Team

Round 2
Reviewer 2 Report
The manuscript has been substantially improved and I appreciate the new figures enclosed. Now, the manuscript is acceptable in the present form.
Author Response
Thank you for your comment. We sincerely appreciate the time you invested to review our manuscript.
Regards
Authors Team
Reviewer 3 Report
Authors should also review some text links, so that the information is more uniform and clear for the reader.
Author Response
Thank you for your suggestion. We have gone through the manuscript comprehensively to review the links and update the information to make the manuscript easily readable. 'Track changes' are used to mark the changes.
Regards
Authors team.